# Formulation of a 3D Printed Biopharmaceutical: The Development of an Alkaline Phosphatase Containing Tablet with Ileo-Colonic Release Profile to Treat Ulcerative Colitis

**DOI:** 10.3390/pharmaceutics14102179

**Published:** 2022-10-13

**Authors:** Khanh T. T. Nguyen, Franca F. M. Heijningen, Daan Zillen, Kjeld J. C. van Bommel, Renz J. van Ee, Henderik W. Frijlink, Wouter L. J. Hinrichs

**Affiliations:** 1Department of Pharmaceutical Technology and Biopharmacy, University of Groningen, 9700 RB Groningen, The Netherlands; 2The Netherlands Organization for Applied Scientific Research (TNO), 5656 AE Eindhoven, The Netherlands

**Keywords:** 3D printing, powder bed printing, personalized medicine, biologics, formulation, ileo-colonic targeting, film coating, ColoPulse, controlled release, ulcerative colitis

## Abstract

Powder bed printing is a 3D-printing process that creates freeform geometries from powders, with increasing traction for personalized medicine potential. Little is known about its applications for biopharmaceuticals. In this study, the production of tablets containing alkaline phosphatase using powder bed printing for the potential treatment of ulcerative colitis (UC) was investigated, as was the coating of these tablets to obtain ileo-colonic targeting. The printing process was studied, revealing line spacing as a critical factor affecting tablet physical properties when using hydroxypropyl cellulose as the binder. Increasing line spacing yielded tablets with higher porosity. The enzymatic activity of alkaline phosphatase (formulated in inulin glass) remained over 95% after 2 weeks of storage at 45 °C. The subsequent application of a colonic targeting coating required a PEG 1500 sub-coating. In vitro release experiments, using a gastrointestinal simulated system, indicated that the desired ileo-colonic release was achieved. Less than 8% of the methylene blue, a release marker, was released in the terminal ileum phase, followed by a fast release in the colon phase. No significant impact from the coating process on the enzymatic activity was found. These tablets are the first to achieve both biopharmaceutical incorporation in powder bed printed tablets and ileo-colonic targeting, thus might be suitable for on-demand patient-centric treatment of UC.

## 1. Introduction

As healthcare is shifting towards personalized medicine, research facilitating this approach is also on the rise. For drug formulation and production, alternative manufacturing methods such as three-dimensional printing (3DP) are gaining increasing attention [1]. three-dimensional printing allows the generation of dosage forms with a wide variety in shape, dosage and release rate such as rapid disintegration, zero-order release and pulsatile release of the active pharmaceutical ingredient (API) [1,2]. Not only does 3DP benefit on-demand production and a personalized medicine approach, but it also facilitates the drug development processes [1]. Unlike 3DP, the conventional manufacturing process (i.e., compaction) is less suitable to provide the customizable dosage and small-scale batches required for clinical trials [1].

3DP currently consists of several techniques, including fused deposition modeling, selective laser sintering and binder jetting or powder bed printing. Each is based on unique core principles, leading to technique-specific advantages and disadvantages [1,3]. With fused deposition modeling and selective laser sintering, in which high temperatures are used, risking drug degradation during the process, especially if the APIs are biopharmaceuticals. Binder jetting, also referred to as powder bed printing, is a form of 3DP during which elevated temperatures are avoided. It is based on selective deposition of liquid droplets on top of an even layer of powder. This powder contains an API, a bulking agent and an excipient that (partially) dissolves in the deposited liquid, often referred to as a binder. The liquid droplets cause the binder to (partially) dissolve after which the powder particles adhere together to form a solidified layer. By stacking multiple layers, followed by a drying step to remove the residual solvent, the desired 3D structure is formed. The printing liquid can also be a mixture of several liquids and can even comprise dissolved materials. The properties of the binder, the liquid and the powder affect the printing process and the properties of the resulting tablets [2]. The two major characteristics of the printing powder are the topology, especially the particle size, and the reactivity to the binder [4,5,6]. Organic solutions of polymers have been used extensively as both a deposition liquid and binder [2]. Formulation of such a solution can be challenging as surface tension and viscosity need to be controlled carefully to ensure printability [7,8]. It is also possible to have a simple deposition liquid having just the solvent while the binder is mixed directly into the printing powder, referred to as solid binder [9]. The powder-binder wettability, liquid droplet penetration and binder spreading behavior are important factors being considered when choosing appropriate materials for powder bed printing [4].

Concurrent with the ongoing emphasis on utilizing proteins and other biological molecules in the pharmaceutical industry, the possibility of making 3D-printed dosage forms containing them is also increasingly being investigated [10]. However, the available studies are still limited to niche applications, such as microneedles [10]. Nevertheless, these studies did not have biopharmaceuticals included in the product directly from the 3D printing but were rather coated on top of the delivery system after the fabrication [10]. Any further application for oral delivery is complicated by the fact that protein molecules often have low bioavailability. So far, other studies on printing tablets containing proteins with direct clinical relevance have yet to be published.

Ulcerative colitis (UC) is a chronic inflammatory bowel disease (IBD) attributed to ongoing and uncontrolled activation of the mucosal immune system [11,12]. The exact causes are multifactorial and a result of a combination of genetic, immunological and bacterial factors. Current treatment is limited to symptom management [13]. Intestinal alkaline phosphatase was shown to play a central role in the regulation of intestinal inflammation and a decreased expression of this protein has been associated with IBD [14,15]. Exogenous intestinal alkaline phosphatase was shown to improve disease activity scores when used to treat IBD in patients who had moderate to severe ulcerative colitis, with no adverse events documented [16]. However, the administration was inconvenient (intraduodenally), resonating with the same difficulty for the administration of other biopharmaceutical treatment options for IBD [12]. For alkaline phosphatase to reach the colon intact, protection against the digestive environment of the upper gastrointestinal tract prior to release in the colon is required.

As shown in several in vitro studies and clinical trials in healthy subjects and Crohn’s disease patients [17,18,19,20,21,22], pH-dependent ColoPulse-coated oral dosage forms can reproducibly provide drug delivery in the ileo-colonic region. The ColoPulse system is a pH-dependent coating based on Eudragit S100, a polymer that dissolves at a pH above 7.0 and contains the non-percolating incorporation of the superdisintegrant AcDiSol. After ingestion, a ColoPulse-coated system will travel unaffectedly through the gastrointestinal tract where the pH stays below 7.0 until the terminal ileum is reached. At this point, the pH increases and eventually exceeds 7.0, leading to the dissolution of Eudragit S100 and rapid disruption of the coating occurs facilitated by the superdisintegrant [23,24,25,26,27]. By supplying a solid dosage form containing alkaline phosphatase with the ColoPulse coating, delivery of this enzyme to the colon might be possible to explore its potential treatment for UC patients.

As the dosage for BIAP in the treatment of UC is not yet established, clinical trials are needed requiring small batches of drugs, and personalized treatment might be recommended. A flexible dosage form that can contains different doses produced on a small scale is desirable. This might be achieved with 3D printing. Therefore, the aim of this study was twofold: to investigate the incorporation of alkaline phosphatase (a therapeutic protein) into a customizable 3DP product made by powder bed printing and to create an ileo-colonic delivery profile for the printed tablets using the ColoPulse coating. The tablets were printed after screening different binders and then characterized. Subsequently, the ColoPulse coating was applied on the 3D-printed tablets, after which the coated tablets were tested in a gastrointestinal simulation system (GISS). During several steps of the process and after storage of the final tablet formulation at elevated temperatures, the enzymatic activity of the protein was assessed.

## 2. Materials and Methods

### 2.1. Materials

Polyvinylpyrrolidone (PVP K30) (Duchefa Farma B.V., Haarlem, The Netherlands), hydroxypropyl cellulose SLFP and LFP (HPC-SLFP and HPC-LFP) were obtained from Nisso HPC (Nisso Chemical Europe GmbH, Düsseldorf, Germany). Bovine intestinal alkaline phosphatase (BIAP), PEG 1500, ammediol, and HEPES were obtained from Sigma-Aldrich (St. Louis, MO, USA). Kollidon VA64 was purchased from BASF Pharma, Ludwigshafen, Germany. Eudragit L100 was obtained from Pharma Excipients, Steinhausen, Switzerland and D-mannitol was purchased from VWR Chemicals BDH, Radnor, PA, USA. The molecular weight (Da) and viscosity (mPa·s) of Kollidon VA64, Eudragit S100, Eudragit L100, HPC-SLFP and HPC-LFP are 45,000–70,000/below 10, 135,000/50–200, 135,000/50–200, 100,000/3.0–5.9 and 140,000/6.0–10.0, respectively. Inulin 4 kDa was a generous gift from Sensus (Roosendaal, The Netherlands). Eudragit S100 was a gift from Evonik Operation GmbH (Essen, Germany). Macrogolum 6000 (PEG6000) and talc were obtained from BUFA (IJsselstein, The Netherlands). Croscarmellose sodium (AcDiSol) was obtained from FMC BioPolymer (Philadelphia, PA, USA).

### 2.2. Spray Drying of Inulin Stabilized BIAP

Inulin 4 kDa was dissolved in 2 mM HEPES buffer pH 7.4 at boiling temperature. After the solution cooled down to room temperature, BIAP was added at a 99:1 ratio of inulin to enzyme (*w*/*w*) to create 5% (*w*/*v*) solutions [28]. Spray drying was performed using a Büchi B-290 mini spray drier, equipped with a high-performance cyclone, a B-296 dehumidifier and a B-295 inert loop (Flawil, Switzerland). Inlet temperature was set at 105 °C, with a feed rate of 3.3 mL/min, 50 mm atomizing air flow and 100% aspirator airflow.

### 2.3. Powder Mixture Preparation

All used binders (i.e., PVP K30, HPV SLFP, Kollidon VA64, Eudragit L100 and HPC-LFP) and mannitol were passed through a 100 µm sieve. Mixtures for the 3D-printing screening experiment (binder and mannitol without spray-dried inulin/BIAP (SD inulin/BIAP)) were blended at 24 rpm for 30 min using a tubular mixer Stuart General Rotator STR4 (Reagecon, Shannon, Ireland) in combination with an STR4/3 drum (Antylia Scientific, Vernon Hills, IL, USA). For mixtures used in printing tablets containing SD inulin/BIAP, an equal amount of mannitol was first gradually added to SD inulin/BIAP in a mixing stainless-steel mortar for careful breaking up of agglomerated SD inulin/BIAP particles and mixing them manually with mannitol. The median particle size difference (63.3 µm for HPC-LFP, 52.7 µm for mannitol and 6.2 µm for SD inulin/BIAP, measured by laser diffraction, details not shown) was the reason for this step. Afterward, this mixture and the remaining amount of the binder/mannitol were added to a tubular mixer for mixing at 24 rpm for 30 min. 

### 2.4. Tablet Design, Powder Bed Printing and Formulation Selection

Formulations were printed with an in-house-built powder bed printer as described previously [29]. The 3D structures for the printing tablets were designed with OpenSCAD (version 2019.05) software and then exported to STL format files. These files were sliced using Simplify3D (version 4.0.1) software to generate G-code instructions for the powder bed printer (PBP). The G-code controls all movements during the printing process including the droplet jetting of the ink, which was ethanol. For the jetting of ink onto the powder bed, a solenoid valve (Lee valve INKA2436510H, orifice diameter of 70 μm and FFKM seal material) (Figure 1) with a drop mass of 11–12 μg binder liquid (ethanol) was used. The printing powder mixture was automatically deposited from a powder depositor onto the print platform and spread out into a layer of consistent thickness using a counter-rotating roller. For all prints, a layer thickness of 0.40 mm was used. In this study, formulation and printing process selection was made in two parts.

Firstly, usable line spacing (LS) values were determined. The LS variable is defined as the distance between the center lines of two printed ink lines. Several single-layer squares measuring 14 × 14 mm each (Figure 1A) were printed with increasing LS values, ranging from 0.20 mm to 0.60 mm. Various combinations of different binders (PVP K30, HPC-SLFP, Kollidon VA64, Eudragit L100 and HPC-LFP) and mannitol as the bulking agent were printed, at three different binder:bulk ratios, namely 2:8, 5:5 and 8:2 (*w*/*w*). The selection of the LS was based on visual inspection of the level of wettability of the square by ethanol, good consolidation of the powder and mechanical strength of the printed squares (they must remain intact when picked up). 

Secondly, the best LS values chosen from the previous experiment were selected to print 3D tablets using a flat face bevel edge tablet design (Figure 1B). The perimeter was printed before the infill pattern was printed. Printed tablets were evaluated, highlighted and categorized into three tiers. Green means successfully printed tablets that did not break when picked up, showing little deformation and complete wetting. Highlighted in yellow are the moderately successfully printed tablets, breaking upon pick up and/or showing some deformation and possibly not equal wetting (hole formation on the top of the tablets). Non-highlighted formulations were unsuccessful, as only unsuitable tablets were produced; these tablets broke immediately and/or were strongly deformed and/or were not equally wetted by the ink.

The binder that produced the most successfully printed tablets was selected for 3D printing with SD inulin/BIAP using the same process. Another printing of single-layer squares measuring 14 × 14 mm was conducted at previously found LS values for this powder mixture having SD inulin/BIAP to test the structure formation. Subsequently, after printing, all the printed products were dried overnight at 50 °C in an oven. The excess powder was brushed off the squares using a simple paintbrush and the finished tablets were stored in a nitrogen gas–filled container at 2–8 °C until further analysis.

### 2.5. Tablet Dimension and Weight

The diameter and thickness of uncoated 3DP tablets containing SD inulin/BIAP were measured using a Digital ABS AOS Caliper (Mitutoyo, Veenendaal, The Netherlands). The weight of the tablets was determined using an AE200 Analytical Balance (Mettler Toledo, OH, USA).

### 2.6. Crushing Strength

The crushing strength of uncoated tablets containing SD inulin/BIAP was measured using the Pharmatron 6D tablet hardness tester (Dr. Schleuniger, Solothurn, Switzerland) similar to the description in British Pharmacopoeia 2022. Tablets were analyzed one by one using the automatic mode, in triplicate.

### 2.7. Friability

Friability was determined with a single blade friability tester, Erweka (Erweka, Hessen, Germany) as described in British Pharmacopoeia 2022. A total of 10 tablets containing SD inulin/BIAP were dedusted, weighed and placed carefully in the machine rotating at 20 rpm for 5 min before being dedusted and weighed again. Measurements were performed in triplicate to calculate the relative weight loss.

### 2.8. Disintegration Time

The disintegration time of tablets was determined using a European Pharmacopoeia 7.0 standard DT2 disintegration tester (Sotax, Aesch, Switzerland) using demineralized water at 37 °C as a medium. Measurements were performed in triplicate.

### 2.9. Scanning Electron Microscopy

Tablet surface morphology was analyzed using a JSM 6460 scanning electron microscope (SEM) (JEOL, Tokyo, Japan) as described previously [30]. Specifically, tablets were fixed on sample stubs by a double-sided adhesive carbon tape before being sputter coated with 10 nm of pure gold using a JFC-1300 auto fine coater (JEOL, Tokyo, Japan) purged in argon gas. Imaging took place under a high vacuum, with a spot size of 25, an acceleration voltage of 10 kV and a working distance of 10 mm.

### 2.10. Solid-State Characterization

Samples were characterized using modulated differential scanning calorimetry (mDSC) and X-ray powder diffraction analysis (XRD) as previously described [28,31]. Tablets were crushed and ground manually into powder prior to analysis. The process of mDSC was performed using a Q2000 supplied by TA instruments (New Castle, PA, USA). Samples of 2 to 3 mg were weighed in aluminum pans, which were subsequently closed. All the pans were loaded into the DSC and stabilized at 20 °C and a temperature ramp from 20 °C to 210 °C was registered at 2 °C/min with 0.318 °C/min modulation. The glass transition temperature was defined as the inflection point of the step transition in the reversing heat flow signal of the thermogram. Samples were measured in duplicate.

XRD was performed using a Bruker D2 Phaser (Billerica, MA, USA). The scanning range was set at 5 to 60° 2θ with a step size of 0.05° 2θ and a step time of 1 s. The detector opening was 5°. The sample stage spun at 15 rounds per minute during measurements. The divergence slit and air scatter screen was 1 mm and 3 mm, respectively. A Si low-background holder was used. Samples were analyzed once.

### 2.11. Enzymatic Assay

Enzymatic activity of BIAP was determined by measuring the enzymatic conversion of a substrate, p-nitrophenyl phosphate (p-NPP) [28,32]. The substrate p-NPP is a colorless compound that is converted to a yellow product that can be detected at 415 nm. To 160 µL of 0.05 M ammediol (pH 9.8) containing 1% MgCl_2_ (*w*/*v*), 20 µL of sample containing 10 µg/mL of protein was added and equilibrated in a stove at 37 °C for 10 min. Subsequently, 20 µL of 5 mg/mL p-NPP was added to this mixture and the absorption at 405 nm was measured every 30 s, for 5 min, in a Synergy HT plate reader (BioTek, Winooski, VT, USA) that was equilibrated at 37 °C. The plate reader was set up to shake the plate for 15 s between each measuring interval. Calibration curves were prepared at BIAP concentrations ranging from 0 to 10 µg/mL, while tablets were dissolved in corresponding amounts of 0.01% BSA in ultrapure water to yield a concentration of 10 µg/mL BIAP. The activity of samples was calculated based on the conversion rate of p-NPP, which was fitted to the calibration curves of the references using linear regression analysis. Measurements were performed in triplicate.

### 2.12. Stability of Alkaline Phosphatase in the Tablet

To assess the impact of the manufacturing process on BIAP stability, the enzymatic activity assay was performed using the printing powder after each processing step. These steps were: after Turbula mixing, during printing and after printing. Measurements were performed in triplicate.

To evaluate storage stability, uncoated tablets containing SD BIAP were stored in an oven at 45°C under low-moisture conditions (<10% relative humidity [RH]) as previously described [28]. The enzymatic activity of BIAP was assessed after 1, 2 and 4 weeks of storage. Additionally, enzymatic activity after the coating process was also evaluated. Measurements were made in triplicate.

### 2.13. Sub-Coating and ColoPulse Coating

The ColoPulse coating consists of Eudragit S100:PEG6000:AcDiSol:Talc in a weight ratio of 7:1:3:2 dissolved or dispersed in ethanol 96% [17]. The 3DP tablets were either directly supplied with the ColoPulse coating or were supplied with a sub-coating comprising approximately 110 mg of PEG 1500 and methylene blue 1% (*w*/*w*) per tablet on average, which was applied using dip coating in melted PEG 1500 prior to application of the ColoPulse coating. Dip coating was performed by submerging the 3D tablet in 70 °C melted PEG 1500 for 5 s before taking it out for solidification. Subsequently, some edges of the tablet were covered by applying melted PEG 1500 manually with a brush. A mini-rotating drum at 32 rpm equipped with a nozzle with a bore diameter of 1 mm (Schlick 970, Düsen-Schlick, Coburg, Germany) driven by a peristaltic pump (Minipuls 3, Gilson, Viliers le Bel, France) at a spray rate of 0.75–1.0 mL/min connected to it was used to apply ColoPulse coating. Dummies were added together with the 3DP tablets during the coating process, up to the total count of 40 tablets in the tumbler. The temperature was maintained within 20–25 °C in the tumbler. Afterward, coated tablets were left drying in the rotating tumbler for 5 min before transferring into an oven for drying at 30 °C for 2 h. Three different coating thicknesses were applied, 15, 17 and 20 mg/cm^2^.

### 2.14. Drug Release Profile Testing in the Gastrointestinal Simulated System

The performance of the coated tablets was tested in a previously described gastrointestinal simulation system [33]. In this system, four different dissolution media are used to simulate the passage through the gastrointestinal tract. The characteristics of the four different phases are presented in Table 1. At the end of each phase, a switch solution was added in 5–10 min using a peristaltic pump to obtain the required composition of the next phase. The composition of each switch solution is provided in Table 2. The tests were carried out in a USP dissolution apparatus type 2 (Sotax AT 7, Sotax, Basel, Switzerland) at 37 °C and a paddle speed of 50 rpm. The release of methylene blue was followed by measuring absorption of the media at 664.5 nm using an in-line UV-spectrophotometer (Evolution 300 UV–VIS spectrophotometer, Thermo Fisher Scientific, Madison, WI, USA), with samples taken every 5 min over 8 h. Measurements were made in triplicate.

### 2.15. Statistical Analysis

Results for experiments were analyzed using statistics that came from a triplicate measurement for each sample in a specific test (except for mDSC, XRD, SEM and drug release) for three independent samples. Data are expressed as means ± standard deviation and analyzed using one-way ANOVA with post hoc Tukey’s multiple comparison test, and results with calculated *p* < 0.05 were considered significantly different.

## 3. Results

### 3.1. Screening of LSs and Binders for the Printing Process

The best LS values for each individual binder that could form acceptable 14 × 14 mm squares with mannitol at different bulk:binder ratios were selected to print tablets for evaluation. The results are categorized as mentioned previously in Section 2.4 and presented in Table 3. HPC-LFP presented the highest count of green and yellow category tablets and thus was chosen as the binder for further experiments on printing tablets containing SD inulin/BIAP. A binder:mannitol ratio of 2:8 (*w*:*w*) was chosen as a basis for SD inulin/BIAP inclusion. This was due to preliminary disintegration tests showing that tablets prepared at a binder:mannitol ratio of 5:5 (*w*:*w*) dissolved much slower (30 min) than tablets prepared at a binder:mannitol ratio of 2:8 (*w*:*w*; 15 min). Tablets made of HPC-LFP: mannitol:inulin/BIAP at a ratio of 2:7.5:0.5 (*w*/*w*/*w*) were printed and studied. The LS values to print these tablets, however, were different in the 14 × 14 mm square printing experiment for this powder mixture. As such, 0.40, 0.45 and 0.50 mm LSs were found to be suitable for printing these tablets with SD inulin/BIAP and thus were selected to print tablets for further experiments.

### 3.2. Three-Dimensional Printed Tablet Properties

The physical properties of printed tablets with the composition HPC-LFP:mannitol:inulin/BIAP 2:7.5:0.5 using three LS values 0.40, 0.45 and 0.50 mm are shown in Table 4. An increased LS resulted in a decrease in average weight, crushing strength, as well as disintegration time of the printed tablets. One interesting observation was that during the crushing strength test, the tablet did not break apart but deformed and got slightly squeezed instead. The lack of residual stress, which is usually present in compressed tablets, might be the cause for this deformation rather than cracking. Obviously, the diameter and thickness of the tablets were independent of the LS. Moreover, SEM (Figure 2) showed that the surface morphology of tablets printed with increased LS displayed a more porous structure. Despite the less porous appearance, friability testing showed that tablets printed with an LS of 0.40 mm had the highest weight loss from the test, while tablets printed at an LS of 0.45 mm showed the least weight loss (although it was not significantly different from 0.50 mm LS tablets).

### 3.3. Solid-State Characteristics of the Tablets

Figure 3 shows the total heat flow DSC thermograms (3A) and XRD patterns (3B) of mannitol, HPC-LFP and the tablets printed at LSs of 0.40 mm, 0.45 mm and 0.50 mm. Figure 3A shows that there is a clear melting endotherm of unprocessed mannitol at around 164–165 °C, coinciding with the melting point observed in the thermograms of 0.40, 0.45 and 0.50 mm LS tablets. XRD analysis (Figure 3B) shows a small amorphous halo covered by many Bragg peaks indicating the presence of crystals in all tablet samples. The HPC-LFP and the SD inulin/BIAP were amorphous (Figure 3B and Figure 4B). The Bragg peaks in the XRD profile of the tablets (Figure 3B) can be ascribed to mannitol.

SD inulin/BIAP showed a clear glass transition temperature at 153.5 °C (Figure 4A); however, it could not be observed in the DSC thermogram of the printed tablet (Figure 3A). This is likely due to the low percentage of SD inulin of just wt-5% in the tablet. To assess the solid state of the SD inulin/BIAP after being exposed to the printing process, an additional experiment was carried out. SD inulin/BIAP was subjected to conditions simulating the printing conditions. I.e., the SD inulin/BIAP powder was left at room temperature (20–25 °C) for 2 h before being wetted by ethanol 100%. Subsequently, the inulin/BIAP sample was put in an oven at 50 °C for over 2 h to allow the ethanol to evaporate, after which it was analyzed by mDSC and XRD. No change in both mDSC and XRD profiles was observed in the SD inulin/BIAP before and after exposure to ethanol (Figure 4).

### 3.4. Enzymatic Activity of Alkaline Phosphatase

Alkaline phosphatase activity was used as an indicator for protein stability and was measured following different steps of the printing process. Only 0.45 mm LS printed tablets were analyzed as they were used for subsequent coating and testing experiments. The enzymatic activity remained the same during the whole printing process (Figure 5), as no significant difference was observed between different steps after printing powder mixing/preparation and 3D printing. Enzymatic activity during these steps was roughly 95%. Some loss of activity found between the original SD powders and the powder after mixing was possibly due to the adhesion of SD inulin/BIAP to various surfaces during the multi-step mixing process.

During storage at 45 °C and low-moisture conditions (<10% RH) of 0.45 mm LS 3D printed tablets containing SD inulin/BIAP, the enzymatic activity was assessed after 1, 2 and 4 weeks. Despite some reduction from week 0 to week 1, storage for up to 2 weeks does not incur any further impact on enzyme stability, with enzymatic activity remaining close to 100% of the theoretical activity (Figure 6). After 4 weeks of storage, the enzymes showed some level of degradation, with enzymatic activity dropping to around 84%.

### 3.5. ColoPulse Coating and In Vitro Release Profile of the 3D Printed Tablets

#### 3.5.1. Coating Process Optimization

To achieve the desired ileo-colonic release profile, it was decided to supply the 3DP tablets with the ColoPulse coating. Due to having good friability and crushing strength, only 0.45 mm LS 3D printed tablets containing SD inulin/BIAP were used for these coating experiments. Applying this coating directly onto these tablets, however, was not possible. Due to their highly porous nature and rough surface (Figure 2), the printed tablets could not be covered completely by the ColoPulse. Despite several attempts to modify the coating process by changing the feed rate, the number of dummies, type of solvent and drying temperature, SEM pictures indicated that even at a large coating thickness (15, 20 and 25 mg/cm^2^), it was impossible to achieve complete and effective coverage of the tablet surface, in particular on the sides of the tablets (Figure 7A–C). This wildly exceeds the original and often used thickness of 8–10 mg/cm^2^ required to achieve ileo-colonic targeting from the application of ColoPulse coating [17]. Direct application of ColoPulse on the 3D printed tablet thus was not feasible. 

To achieve complete coverage of the tablet with the ColoPulse coating, a sub-coating consisting of PEG 1500 was applied. An SEM image of the tablet after sub-coating with PEG 1500 is shown in Figure 8B. Compared with tablets without sub-coating (also Figure 8B), the PEG 1500 coated tablets have a smooth surface. After coating these sub-coated tablets with the ColoPulse, the smoothness decreased (Figure 8B). The roughness of the ColoPulse coating was likely primarily due to AcDiSol particles, with some contribution from talc, as they do not dissolve in the coating suspension.

The whole coating process did not lead to any significant reduction in enzymatic activity of alkaline phosphatase when comparing uncoated tablets and tablets coated with both PEG 1500 and ColoPulse (Figure 9). The enzymatic activity of tablets after PEG 1500 coating was slightly but significantly lowered compared with before coating, but the activity returned to normal in tablets with both the PEG 1500 and ColoPulse coating.

#### 3.5.2. In Vitro Release of Methylene Blue from ColoPulse-Coated 3D Printed Tablets

The release of the ColoPulse-coated tablets was tested in an in vitro simulation system consisting of 4 phases, with phase 4 representing the colon. The ColoPulse coating is designed to open in phase 3 (simulating the terminal ileum), leading to the dissolution and release of the alkaline phosphatase. The release of alkaline phosphatase was not monitored directly because the concentration of BIAP in GISS was below the lower limit of detection for the enzymatic activity assay. Instead, to monitor the release, methylene blue was added to the molten PEG 1500 at a 1% *w*/*w* concentration prior to dip coating. As methylene blue can be observed visually, premature release can easily be detected by visual observation during preliminary experiments. As PEG 1500 is highly soluble in water and the tablet disintegrates fast (Table 4), it was assumed that the release of the dye closely corresponds to the release of the protein from the printed tablet. As seen in Figure 10, there was almost no methylene blue detected in the dissolution media simulating the gastric and intestinal phases in the first 200 min. Close to the end of phase 2, there was only 8% and 3% of methylene blue release detected for the tablet coated at 15 and 17 mg/cm^2^ with the ColoPulse system, respectively. For the ColoPulse system, the standard requirement is to have less than 10% drug release after 240 min in the GISS (the end of simulated jejunum and start of simulated ileum). However, tablets supplied with 17 mg/cm^2^ coating showed incomplete release of methylene blue in phases 3 and 4, as the coating did not rupture completely, reaching not more than around 40% release. In contrast, tablets supplied with 15 mg/cm^2^ coating reached 89% release on average, and both the coating and the core disintegrated completely. The impact of the ColoPulse coating thickness was further seen for the tablets supplied with a 20 mg/cm^2^ coating (Figure 10) as there was little to no methylene blue release in the first 4 h and a constant slow release afterward.

## 4. Discussion

In this study, the incorporation of BIAP into powder bed printed tablets was examined and an ileo-colonic delivery coating was applied to potentially treat UC. The choice of LSs was specific for the different binders used, with HPC-LFP being the most flexible binder to form 3D-printed tablets. Inulin was shown to be an excellent stabilizer for BIAP, retaining both the amorphous state as well as enzymatic activity throughout the production process. This is in line with previous findings of inulin’s excellent protein stabilization capability [28,34,35,36]. Ileo-colonic targeting was achieved by applying the ColoPulse coating with a PEG 1500 sub-coating on the 3DP tablets. Using the in vitro GISS test, the release profile of the tablets was evaluated showing the desired ileo-colonic targeting pattern.

Of all the binders screened for powder bed printing, HPC-LFP demonstrated a strong capability to form a tablet at different ratios of binder:mannitol as well as LS values, possibly due to its good binding characteristics and wetting potential by ethanol. This is similar to a previous study that showed the flexibility of different grades of HPC-LFP as a solid binder for powder bed 3D printing when the ink consists of only solvents [9]. The printed tablets had a porous structure, typical for products made from powder bed printing [37,38]. There is a good correlation between several physical properties of the tablets (crushing strength, dissolution time and weight) and the LS values used in the printing process. As lower LS values correspond to higher liquid amounts, this might result in more consolidation of powder particles, in turn resulting in a solid structure with better physical properties (Figure 2), similar to the results observed in another study [29]. There is a difference between tablets printed at an LS of 0.50 mm and those printed at 0.40 or 0.45 mm (although not significant with 0.45 mm LS tablets). The tablets prepared at an LS of 0.50 mm disintegrated faster and had lower crushing strengths. Friability values are slightly above the 1% limit of most pharmacopoeias. While two other papers reported their tablets to have friability below 1%, in these studies the binders were dissolved in the printing liquid, which may have led to stronger tablets [39,40]. In another study where the binder was mixed with the printing powder, the friability of the tablets was reported to be around 1–3%, which is comparable with the result of this study [9]. As tablets were dried until no weight change could be detected, most residual solvents had been removed from the product. Additionally, according to the FDA and the EMA, an intake of 50 mg of ethanol per day is acceptable. This is significantly higher than what can possibly be present in a tablet of 115 mg before coating (Table 4). Therefore, the risk of toxicity from residual solvent in these tablets is minimal.

The enzymatic activity of BIAP was reduced after mixing but remained unchanged during the printing process (Figure 5). After 2 weeks of storage at 45 °C, the enzyme remained intact despite the high-stress conditions. Although not significantly different, activity was at a loss after 4 weeks. The coating process did not introduce any stress to BIAP in the final PEG sub-coated and ColoPulse-coated tablet, as seen in Figure 9. In a previous study where infliximab-containing tablets were coated, a certain degree of protein damage was observed when using acetone as the medium for the ColoPulse suspension [12]. Although the exposure of BIAP to ethanol may not have the same negative consequence, the PEG 1500 layer might have helped to protect the enzyme from possible degradation caused by the exposure to ethanol during the application of ColoPulse coating. 

Due to the rough surface of powder bed printed tablets, it was impossible to directly apply a closed and continuous layer of ColoPulse coating, as can be seen in Figure 7. The rough surface of uncoated tablets probably leads to unbalanced deposition of the ColoPulse and incomplete coverage of the tablet surface. Many holes could be observed on the sides of the tablets, comprising locations lacking the coating, while other sections might have received a coating that was thicker than desired, leading to issues with the dissolution of the coating on those sections. Therefore, to allow the application of a functional ColoPulse coating, a sub-coating consisting of approximately 110 mg PEG 1500 was applied by dip coating at 70 °C. To obtain a tablet with an acceptably smooth surface, a relatively high amount of PEG 1500 had to be used as sub-coating. This can be attributed to the porous structure of the tablet. This porous structure results in the absorption of PEG 1500 into the core of the tablet by capillary forces. This might delay the release of the drug from the core tablet. Nonetheless, from preliminary testing, this delay is limited to only 10 to 15 min (data not shown). Figure 8 shows the SEM images of the PEG 1500 sub-coated tablets and the tablet with both the PEG 1500 and ColoPulse coating. The surface of the tablets after PEG 1500 sub-coating was significantly smoother than that of uncoated tablets (Figure 8). Furthermore, the sub-coating also protected the tablets from abrasion during the ColoPulse coating process, compensating for the slightly higher friability of the 3DP tablets than the pharmacopoeial limit of 1%. Subsequent ColoPulse application created a slightly rough yet still continuous layer over the PEG 1500 layer. In previous studies, binder-jet 3D printing has been employed to create dosage forms with complex release profiles, such as zero-order release or a core-shell structure [41,42]. Fabricating a core-shell construct for controlled release often requires the API to be dissolved in the liquid for selective deposition in the core region of the printing powder [40,42,43,44]. This may cause stability issues, especially for biopharmaceuticals as the liquids used are mostly organic solvents. As a result, to the best of our knowledge, a targeted and controlled release system for biopharmaceuticals using PBP systems has not been realized yet. There is no study exploring the coating of a pH-dependent release layer on top of powder bed printed tablets. The rough and inhomogeneous surface of uncoated tablets presents a challenge for the direct application of different controlled-released coatings. In one study the authors applied an enteric coating layer directly on top of a tablet produced by fused deposition modeling, but the surface of these tablets was smoother than the powder bed printed tablets in this study, making such a direct application possible [45].

The pH peak at 7.4 in the terminal ileum forms the basis of the ColoPulse pH-dependent release system [17,18,19,46,47,48,49]. Different setups mimicking the precise phases of the gastrointestinal tract can be found in the review by Broesder et al. [23]. Due to the opening mechanism of the ColoPulse, which is the rupturing of the coating at a pH above 7.0, a testing condition mimicking this specific condition is required, such as the GISS. This is further explained in [33]. The release profile of methylene blue was used to examine the release kinetics of tablets coated with the ColoPulse coating. The level of coating applied on the 3D printed tablets strongly affected the tablet release characteristics and was found to be critical to the release performance. At a coating level of 15 mg/cm^2^, some early release in the first 4 h was seen, whereas at higher coating levels (17 and 20 mg/cm^2^), this early release was almost completely absent until the GISS reached a pH 7.6 phase (Figure 10). The thickness of ColoPulse coating required on the 3D printed tablets was considerably higher than previously described for tablets or gelatin capsules, where only 8-12 mg/cm^2^ was needed [17,19]. In preliminary experiments, it was visually observed that ColoPulse coating thicknesses below 15 mg/cm^2^ resulted in premature methylene blue release (data not shown). Since the ColoPulse coating suspension is ethanol based, some PEG 1500 could be dissolved during coating, possibly influencing the properties of the ColoPulse layer. The slow almost zero-order release of tablets supplied with 17 and 20 mg/cm^2^ of ColoPulse coating in phase 4 could be the result of two major factors. Firstly, the ColoPulse coating did not rupture significantly in phase 3 where pH was above 7.0. Secondly, when water penetrated the core through these small ruptures, the core could form a gel due to the presence of HPC-LFP, as seen by cutting the tablet in half after the dissolution test (data not shown). Both phenomena, i.e., partial rupturing of the ColoPulse coating and gel formation, may have led to slow diffusion of methylene blue from the core to the dissolution medium. Tablet supplied with 15 mg/cm^2^ did not suffer these issues as the ColoPulse shell completely disintegrated and separated from the tablet core after phase 3. This allowed the contents of the core to freely diffuse into the dissolution medium. The therapeutic dosage of BIAP for UC treatment is not yet established. One study suggested that 30,000 U would be needed [16]. This BIAP load in the 3DP tablets can be achieved in different ways: (1) using pure BIAP grade, (2) increasing the percentage of BIAP in the spray-dried powder, (3) increasing the percentage of SD powder in the 3DP tablet and, finally, (4) increasing the tablet weight. A higher load of BIAP (up to 10%) in the spray-dried inulin matrix can be achieved [28,50]. Pure BIAP can reach activity above 5700 U/mg (provided by Sigma-Aldrich, product number P0114), and thus only 5.2 mg BIAP from this product is required. The presence of stabilized SD inulin/BIAP can be increased from 5% to 30% by replacing the bulking agent mannitol with SD powder while maintaining the relative amount of binder material. As the binder is mainly responsible for the printing behavior, this is expected not to affect the printing process significantly. The tablet dimension, both diameter and thickness, can be increased by 10%, from 8.7 and 3.5 mm to 9.6 and 3.9 mm, respectively. This increases the volume and thus the total tablet weight by approximately 35%, while not dramatically changing the tablet size. Additionally, carrying out printing at lower LS (0.40 rather than 0.45 mm) also produces heavier, denser tablets (increasing the weight from 115 to 130 mg) while maintaining the same dimension (Table 4). The combination of these approaches can achieve the suggested therapeutic dose of 30,000 U. Further increase in the drug load might be possible. While rising the percentage of SD inulin/BIAP in the tablet further from 30% to 40% is within reach, increasing the tablet size may depend on patient acceptance. The percentage of protein in the SD powder can be enhanced, likely up to 20% or beyond, as seen in other studies on stabilizing proteins with other saccharides [51,52,53]. Nevertheless, further investigation as well as adjustment of the sugar glass matrix will be required. These changes may also affect the dissolution of the core and thus studies may be needed to evaluate this effect.

## 5. Conclusions

This study is, to the best of our knowledge, the first to show that it is possible to incorporate biopharmaceuticals in powder bed printed tablets with an ileo-colonic controlled release profile for a potentially clinically relevant application. Physical properties of powder bed 3D printed tablets such as disintegration time are substantially affected by the LS values and thus can be adjusted by changing this parameter. Alkaline phosphatase incorporated in inulin by spray drying remained stable during the printing process and under subsequent accelerated storage conditions. The application of the proprietary ColoPulse coating was made possible by supplying the rough surface of the tablets with a sub-coating consisting of PEG 1500 to finally achieve an ileo-colonic targeted dosage form. In vivo, this might allow alkaline phosphatase to be delivered selectively to the colon, avoiding degradation from the gastric and intestinal environment, thus enabling the possible application in the treatment of ulcerative colitis. The dosage of alkaline phosphatase can potentially be adjusted on demand to produce small custom batches for clinical research as well as personalized medicine for patient-specific treatment. In vivo experiments will be required for further confirmation of treatment effectiveness. Despite overcoming the challenges of creating an ileo-colonic targeting 3DP product, the process of applying the ColoPulse coating on top of a 3D printed tablet is complex and alternative methods to create 3DP tablets with a smoother surface should be explored.

## Figures and Tables

**Figure 1 pharmaceutics-14-02179-f001:**
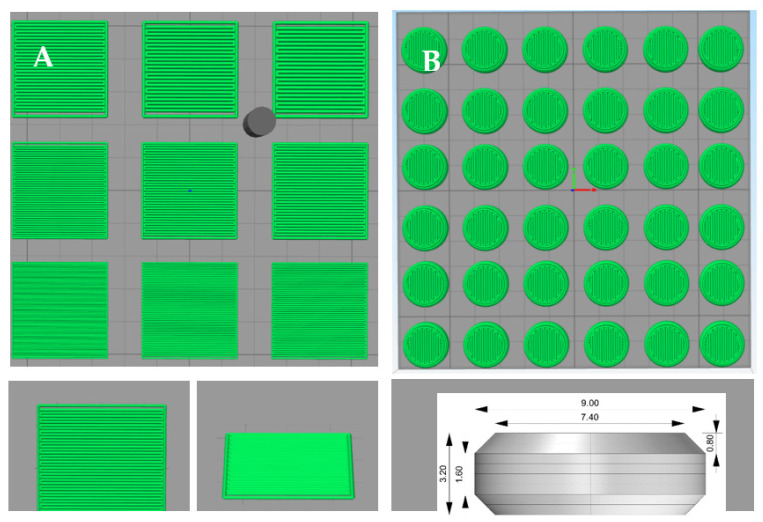
Pattern of droplet deposition: 3 × 3 square design with top and side view single square, height 0.40 mm, sides 14.00 mm (**A**) and flat face bevel edges tablet design, height 3.20 mm, diameter 9.00 mm (**B**).

**Figure 2 pharmaceutics-14-02179-f002:**
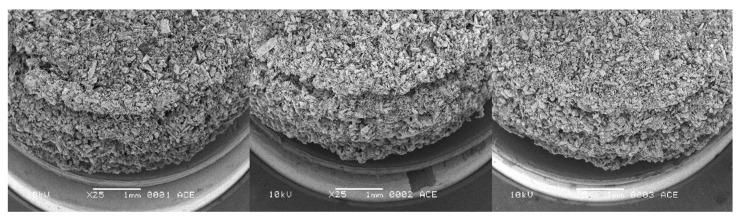
SEM micrographs of tablets printed at different LSs. (**Left**): 0.5 mm, (**middle**): 0.45 mm, (**right**): 0.4 mm. Pictures were taken with a spot size of 25, an acceleration voltage of 10 kV, a working distance of 10 mm, 20° tilt and 25× magnification.

**Figure 3 pharmaceutics-14-02179-f003:**
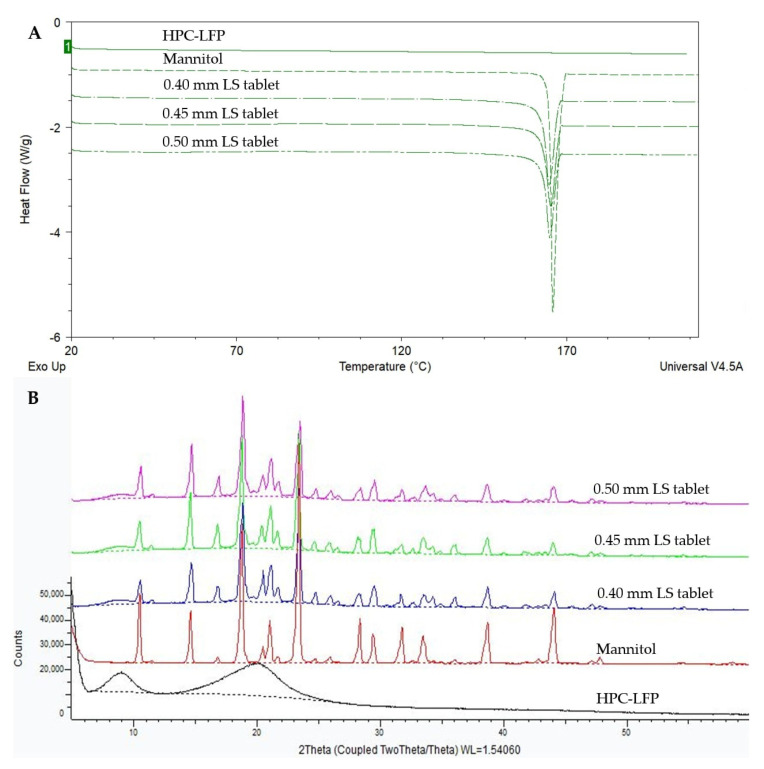
The mDSC thermogram (**A**) and XRD spectra (**B**) of mannitol, HPC-LFP and 3D printed tablets prepared at different LS values.

**Figure 4 pharmaceutics-14-02179-f004:**
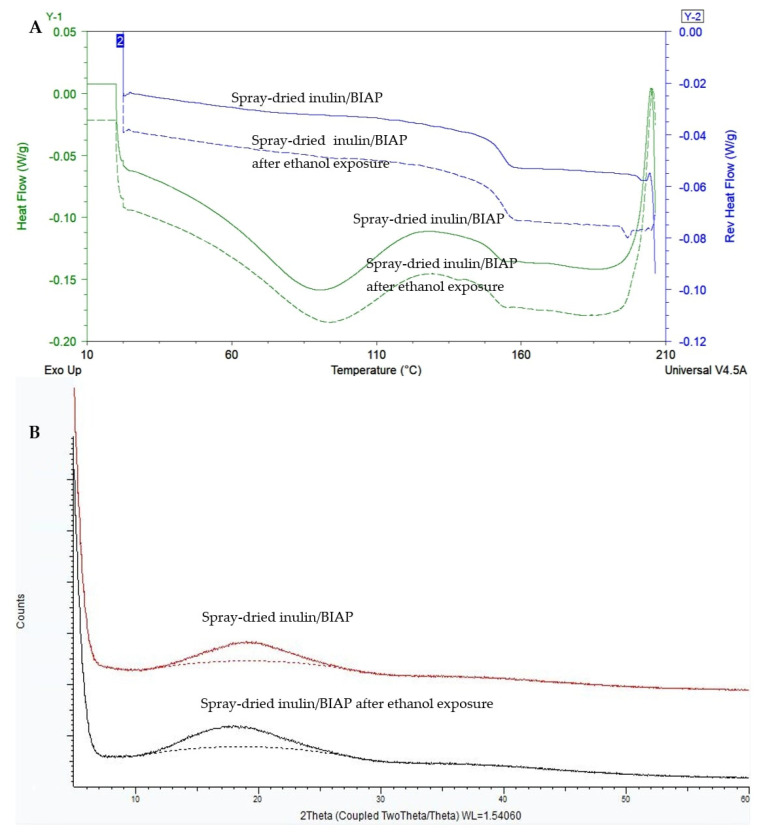
The mDSC thermogram (**A**) and XRD spectra (**B**) of SD inulin/BIAP before and after simulation experiment (storage for 2 h at room temperature (20–25 °C), followed by ethanol wetting and finally drying for 2 h in the oven at 50 °C).

**Figure 5 pharmaceutics-14-02179-f005:**
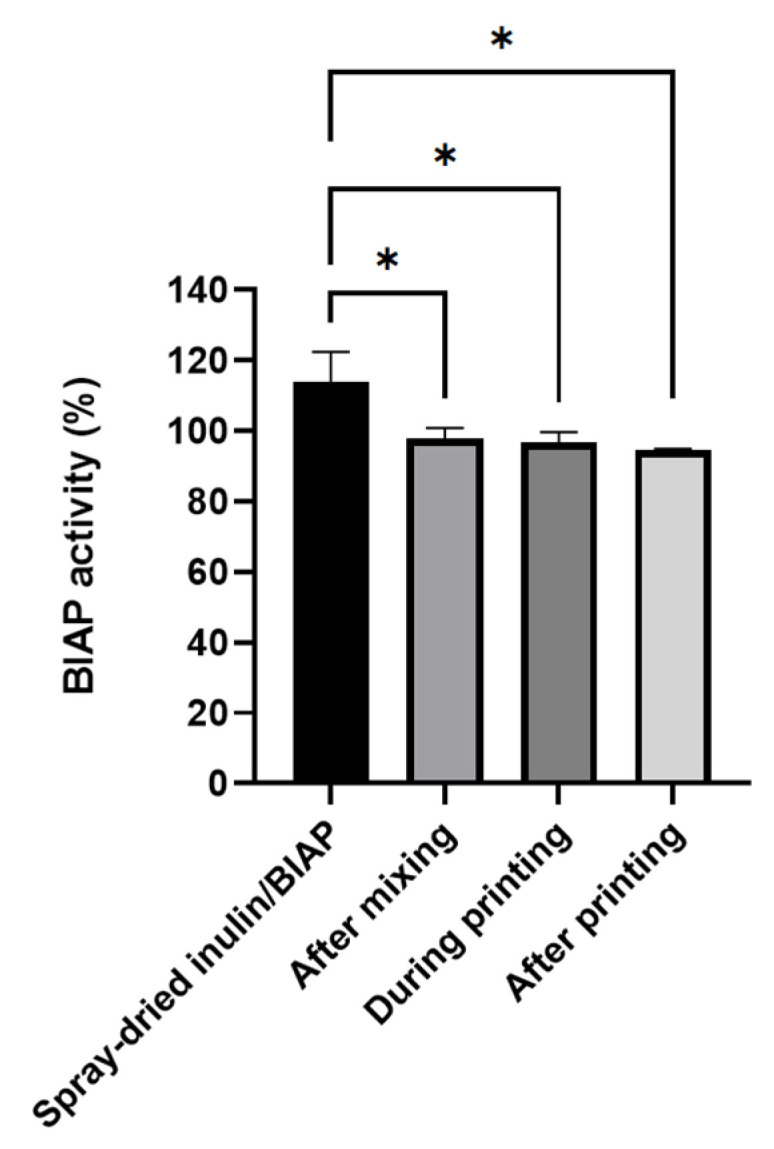
Enzymatic activity of alkaline phosphatase for the preparation of 0.45 mm LS printed tablets after each printing step (n = 3, * *p* < 0.05).

**Figure 6 pharmaceutics-14-02179-f006:**
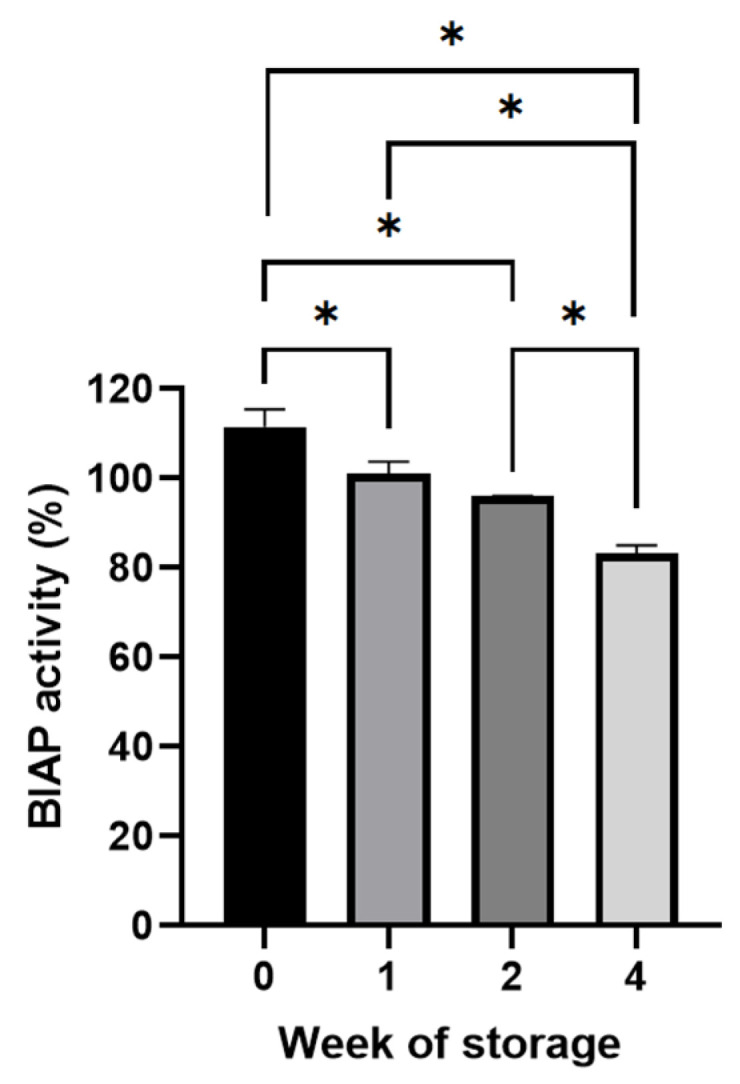
Enzymatic activity of alkaline phosphatase after storage for 1, 2 and 4 weeks at 45 °C of 0.45 mm LS printed tablets (n = 3, * *p* < 0.05).

**Figure 7 pharmaceutics-14-02179-f007:**
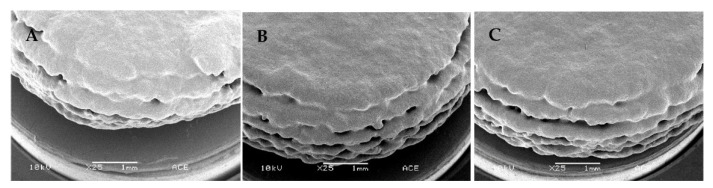
SEM pictures of 0.45 mm LS coated 3D printed tablets with ColoPulse coating thicknesses of (**A**) 15 mg/cm^2^, (**B**) 20 mg/cm^2^ and (**C**) 25 mg/cm^2^. Pictures were taken with a spot size of 25, an acceleration voltage of 10 kV, a working distance of 10 mm, 20° tilt and 25× magnification.

**Figure 8 pharmaceutics-14-02179-f008:**
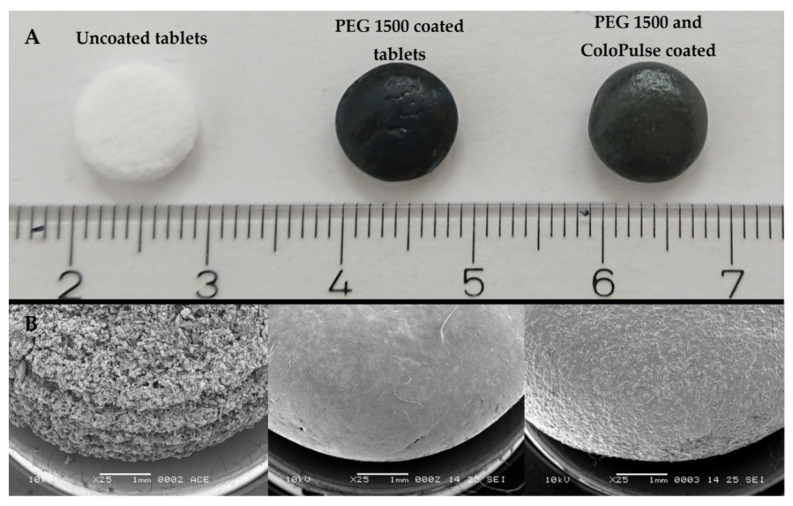
Photograph (**A**) and SEM images (**B**) of 0.45 mm LS tablets, including uncoated tablets, PEG 1500 coated tablets and PEG 1500 with ColoPulse-coated tablets. Pictures were taken with a spot size of 25, an acceleration voltage of 10 kV, a working distance of 10 mm, 20° tilt and 25× magnification.

**Figure 9 pharmaceutics-14-02179-f009:**
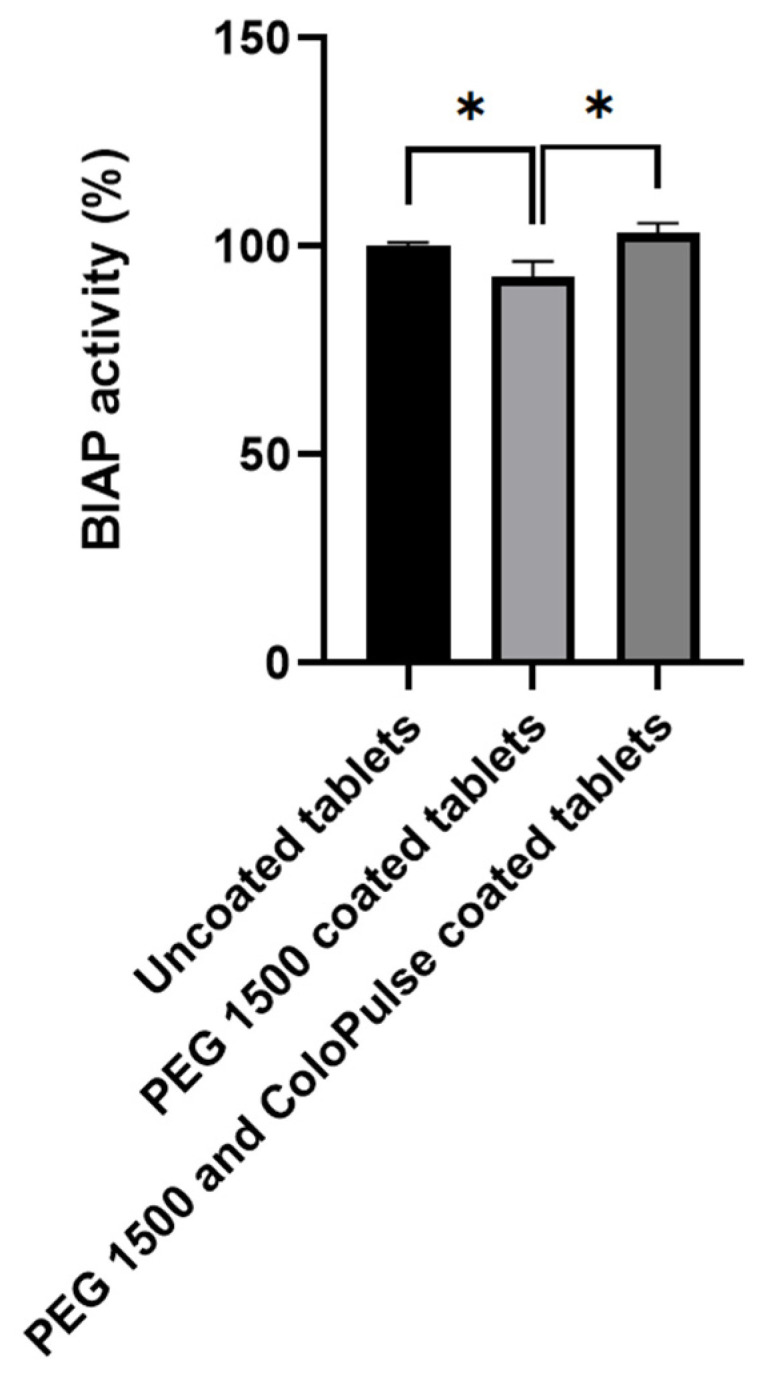
Enzymatic activity of alkaline phosphatase after the coating process of 0.45 mm LS printed tablets. Data were normalized with the BIAP activity in tablets prior to coating (n = 3, * *p* < 0.05).

**Figure 10 pharmaceutics-14-02179-f010:**
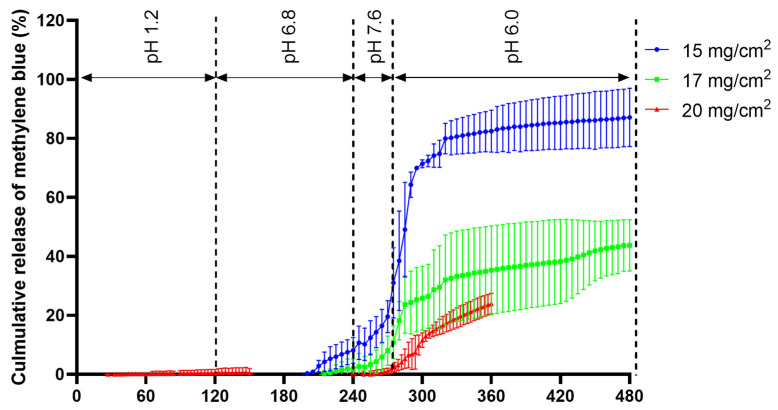
Release profile of methylene blue from 0.45 mm LS 3D printed tablet sub-coated with PEG 1500 and coated with 15, 17 and 20 mg/cm^2^ ColoPulse (n = 3).

**Table 1 pharmaceutics-14-02179-t001:** Specifications of the GISS, adapted with permission from [33].

Phase	Segment Gastrointestinal Tract	pH	Volume (mL)	Time (h)
I	Stomach	1.20 ± 0.20	500	2.0
II	Jejunum	6.80 ± 0.20	629	2.0
III	Terminal ileum	7.63 ± 0.12	940	0.5
IV	Colon	6.00 ± 0.25	1000	1.5

**Table 2 pharmaceutics-14-02179-t002:** Composition of the switch solutions, according to reference, adapted with permission from [33].

	Composition	Time Added to the Dissolution Vessel (h)
Phase I	1.00 g sodium chloride, 3.5 mL concentrated hydrochloric acid, add demineralized water to 500 ml	0
Phase I to phase II	4.08 potassium dihydrogen phosphate, 30 mL sodium hydroxide 2.0 M (80 g/L), add demineralized water to 129 mL	2.0
Phase II to phase III	2.04 g potassium dihydrogen phosphate, 12.0 mL sodium hydroxide 2.0 M (80 g/L), add demineralized water to 311 mL	4.0
Phase III to phase IV	9 mL hydrochloric acid 3.0 M, add demineralized water of 60 mL	4.5

**Table 3 pharmaceutics-14-02179-t003:** List of formulations being screened (without SD inulin/BIAP). Green means successfully printed tablets that did not break when picked up, showing little deformation and complete wetting. Yellow are the moderately successfully printed tablets, breaking upon pick up and/or showing some deformation and possibly not equal wetting (hole formation on the top of the tablets). Non-highlighted formulations were unsuccessful, as only unsuitable tablets were produced; these tablets broke immediately and/or were strongly deformed and/or were not equally wetted by the ink.

Binder	LS (mm)	Binder:Mannitol Ratio (*w*:*w*)		
PVP K30	0.40	2:8	5:5	8:2
	0.45	2:8	5:5	8:2
	0.50	2:8	5:5	8:2
HPC-SLFP	0.45	2:8	5:5	8:2
	0.50	2:8	5:5	8:2
	0.55	2:8	5:5	8:2
Kollidon VA64	0.40	2:8		8:2
	0.45	2:8		8:2
	0.50	2:8		8:2
Eudragit L100	0.35	2:8	5:5	8:2
	0.40	2:8	5:5	8:2
	0.45	2:8		8:2
HPC-LFP	0.50	2:8	5:5	8:2
	0.55	2:8	5:5	8:2
	0.60	2:8	5:5	8:2

**Table 4 pharmaceutics-14-02179-t004:** Physical properties of the HPC-LFP:mannitol:inulin/BIAP tablets (2:7.5:0.5 *w*:*w*:*w*) tablets 3D printed using different LS values (n = 3, ^†^
*p* < 0.05 for pair comparison between 0.40 and 0.45 mm LS, ^∇^ *p* < 0.05 for pair comparison between 0.50 and 0.40 mm).

	0.40 mm LS	0.45 mm LS	0.50 mm LS
Average weight (mg)	132.1 ± 7.4	115.9 ± 5.7	106.7 ± 5.5
Diameter (mm)	8.71 ± 0.09	8.66 ± 0.07	8.62 ± 0.02
Thickness (mm)	3.76 ± 0.21	3.46 ± 0.17	3.35 ± 0.22
Crushing strength (N)	67.0 ± 14.0	64.3 ± 4.6	39.0 ± 7.0 ^∇^
Friability (percentage weight loss)	2.58 ± 0.51 ^†^	1.19 ± 0.26	1.41 ± 0.39 ^∇^
Disintegration time (minutes)	12.26 ± 2.07 ^†^	5.95 ± 0.52	3.36 ± 0.95 ^∇^

## Data Availability

Not applicable.

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
