# Peer review of "Formulation of a 3D Printed Biopharmaceutical: The Development of an Alkaline Phosphatase Containing Tablet with Ileo-Colonic Release Profile to Treat Ulcerative Colitis"

_pharmaceutics, 2022, doi:10.3390/pharmaceutics14102179_

Round 1

Reviewer 1 Report

The manuscript can be accepted after several issues are addressed:

- The binder jetting technology can be extended in the Introduction sestion (the purpose/role/influence of the binding component within the process, etc.)

- The drug release regarding the drug release profile testing in the gastrointestinal part should have a longer release time in the colon area. Please check and provide several references about the setup times for the digestive system.

- The role of the tablets coating process can be better highlighted in the Results section.

- Several SEM images can be added with tablets after the in vitro testing in order to reveal the surface morphology.

Author Response

I have revised the manuscript and thus provide response to your comments in the attachment.

Reviewer 2 Report

The manuscript entitled “Formulation of a 3D printed biopharmaceutical: the development of an alkaline phosphatase containing tablet with ileo-colonic release profile to treat ulcerative colitis” describes the effort to develop a powder bed printing process for an enzyme containing tablet with subsequent traditional tablet coating to achieve colonic delivery. The manuscript is well written and many aspects comprehensive. I have the following comments. 

-       In the introduction, please specify why this dosage form should be 3d-printed opposed to “traditional” manufacturing techniques.

-       Line 75 “After ingestion, a ColoPulse coated system will travel unaffectedly through the gastrointestinal tract where the pH stays below 7.0, until the terminal ileum is reached. At this point, the pH exceeds 7.0, leading to the dissolution of Eudragit S100 and a rapid disruption of the coating occurs facilitated by the superdisintegrant” Please add literature regarding these sharp pH shifts during GI transit or amend your comment. 

-       Line 118 “For mixtures used in printing tablets containing SD inulin/BIAP, an equal amount mannitol was first gradually added to SD inulin/BIAP in a stainless-steel mortar for manual mixing.” Please state whether this was a grinding step (particle size reduction) or “careful mixing”. What was the particle size of your spray dried API with respect to the binder and filler?

-       Line 123 “Formulations were printed with an in-house built powder bed printer as described 124 previously [22]. “ In the cited literature a full name of the printer is given – it seems to be a commercially available system. If so, please clarify.

-       Line 144 “Secondly, the best LS values chosen from the previous experiment were selected to print 3D tablets using a biconvex tablet design (Figure 1B).” Is this truly a biconvex tablet? The image looks like a cylinder. 

-       Did you consider residual solvents in your product? Might this be a problem?

-       Please very briefly describe the setup of the GISS – the cited literature is not freely accessible.

-       Please add photographs of the printed uncoated and coated tablets.

-       Please explain how the dip coating was performed. Please comment on the fact that the mass added during dip coating was almost as high as the tablet mass. Does this influence release?

-       How was methylene blue detected? Why wasn´t the model substance included in the core? Why was this molecule chosen and not a drug? Does the alkaline phosphatase activity determined allow for any conclusions regarding the drug content of the printed tablets and whether this was reproducible?

-       The friability was out of the pharmacopeial range – did you observe any problems during the coating process?

-       Figure 10 – why is there release detectable during the first 150 min for the 20 mg/cm2 coating but no longer afterwards (only after 240 min there is again data for the red marked 20 mg/cm2 coating)? How can the coating not open completely for the 17 and 20 mg/cm2 and only partially release at the end of the experiment? Even if only part was dissolved, the fast disintegrating core should be gone fast and then the peg subcoat should release the methylene blue. The results for release of methylene blue are given as percentage – is this with reference to a theoretical methyelene blue content or to the actually detected amount?

-       Line 571 “The thickness of ColoPulse coating required on the 3D printed tablets was considerably higher than previously described for tablets or gelatin capsules, where only 8-12 mg/cm2 was needed.” How was this determined? Were smaller coating amounts applied and the coating found to release too early in preliminary experiments? Do you have any explanation for this?

-       Line 588 “The dosage of alkaline phosphatase can potentially be adjusted on demand, as tablets can easily be made to any size and thus active pharmaceutical ingredient load, to produce small custom batches for clinical research as well as personalized medicine for patient specific treatment.” Was the dose chosen realistic? Could you dramatically increase the drug load? Due to need to swallow the intact tablet surely tablets cannot be manufactured at any size, even though this is technically feasible…. Please amend this and comment on the dose. 

-       Please check whether all self-citations are really useful. 

Author Response

(The authors gave the same response as above.)

Reviewer 3 Report

In this manuscript, the authors have adopted 3D powder bed printing to formulate a tablet with a desired pH window of ileo-co-lonic release profile to treat ulcerative colitis.  The research objectives are clearly stated, the study was thoroughly conducted however some sections lack specificity and additional details are required for clarity. 

1) The abstract contains only a portion of the significant results not capturing the morphology of the 3D-printed formulations, methylene blue release profiles nor the statistical levels of significance of each test.

2) The physicochemical/rheological characteristic of the materials (MW and viscosity) should be outlined as much as possible . Manufacturer part numbers are not sufficient.

3) Methods

There are no references to literature whether peer-review or industry standards for sections 2.2, 2.3. 2.6,2.7,2.8,2.9,2.10,2.11 and 2.12. Sources should be added or if this is a new method adopted by authors it should be acknowledged.

4) Results

4a) Statistical analysis

It is suggested not to change the significance level for paired comparisons especially since the order of the magnitude of the measurements are the same.  Specifically, in Figures 5 and 6 the pairwise comparisons are meaningless by lowering the p-values since the order or magnitude of the metric is not changing. The evaluation should be conducted at either p<0.05 or p<0.01 for consistency.

For the crushing strength it is stated that for the LS 0.5 mm is not significantly different than the 0.4 mm LS at p< 0.05.  Is the caption for Table 3 correct? Why is number of asterisks changing for the same significance level between LS 0.4-0.5 mm. 

4b) The color code in Table 2 is not clear.

4c) The SEM pictures are very clear but not the associated legend

4d) Dimensions on Figure 1 (not just the caption) closer to a CAD drawing would be helpful.

5) Conclusion

Without the references needed in the methods and additional comparisons in the discussion this statement cannot be validated.

"This study is, to the best of our knowledge, the first to show that it is possible to incorporate biopharmaceuticals in powder bed printed tablets with a clinically relevant application."

Author Response

(The authors gave the same response as above.)

Round 2

Reviewer 2 Report

To make it easier I have just added my comments in red to your rebuttal letter. Please see file attached. 

Author Response

I have added my response to the rebuttal below the previous comments in the same response file for ease of tracing back. Please find them in the attachement.

Reviewer 3 Report

Thank you for addressing all the comments. 

The quality of the submission has significantly increased as well as the interest to the readers.

Author Response

Many thanks for your insightful comments.

Kind regards.